# Birth Control Use and Access Including Pharmacist-Prescribed Contraception Services during COVID-19

**DOI:** 10.3390/pharmacy10060142

**Published:** 2022-10-28

**Authors:** Karli Pelaccio, David Bright, Heather Dillaway, Mary Beth O’Connell

**Affiliations:** 1Doctor of Pharmacy Program, Eugene Applebaum College of Pharmacy and Health Sciences, Wayne State University, 259 Mack Ave., Detroit, MI 48201, USA; 2Pharmaceutical Sciences Department, College of Pharmacy, Ferris State University, 202C Hagerman Pharmacy Building, 220 Ferris Dr., Big Rapids, MI 49307, USA; 3College of Arts and Sciences, Illinois State University, Stevenson Hall 141, Campus Box 4100, Normal, IL 61790, USA; 4Pharmacy Practice Department, Eugene Applebaum College of Pharmacy and Health Sciences, Wayne State University, 259 Mack Ave., Suite 2124, Detroit, MI 48201, USA

**Keywords:** birth control, community pharmacy, COVID-19, hormonal contraception, pharmacy access, pharmacist prescribing

## Abstract

The COVID-19 pandemic influenced health care with effects on contraception access emerging. The study objectives were to analyze pandemic impacts on birth control (BC) use and access; and evaluate perceptions of pharmacist-prescribed BC. A 50-item survey was distributed by 31 Michigan community pharmacies to women aged 18–45 over a three-month period. The survey link was also posted on two websites. 147 surveys were analyzed. Respondents were 29 ± 7.9 years old, primarily white (77%) and straight (81%). Fifty-eight percent of respondents used prescription BC, mostly to prevent pregnancy (84%) with oral pills (76%) being the most common formulation. Some BC users (25%) were worried about BC access and 27% had difficulty taking BC regularly. Half of the respondents (50%) would likely use pharmacist-prescribed BC if available, with advantages being more convenient than visiting a doctor’s office (71%) and easier access (69%). The major concern about pharmacist-prescribed BC was women not receiving PAP smears and screenings (61%). Respondents reported high confidence (72%) in pharmacist-prescribed BC and believe it would help prevent unintended pregnancies (69%). Some respondents experienced altered BC use and access. Half of the respondents supported pharmacist-prescribed BC. Pharmacist-prescribed BC could help solve pandemic-related access problems.

## 1. Introduction

Contraception is essential to maintain women’s health for pregnancy prevention and certain health conditions [1]. In the United States, 45% of pregnancies are unintended [2], which negatively impact outcomes for both mother and child, especially for historically excluded and underserved women [3,4]. Unintended pregnancies impact society, costing the public $21 billion [5]. Effective, convenient, and affordable contraception, along with accessible reproductive health services, are effective ways to prevent unintended pregnancies and reduce healthcare costs [6]. Additionally, contraception treats polycystic ovarian syndrome, menorrhagia, endometriosis, and dysmenorrhea [7]. Women experience many contraception access barriers including cost, insurance coverage, privacy, transportation, lack of full women’s health screenings, and insufficient knowledge of contraception methods and where to obtain birth control (BC) [4,6].

The COVID-19 pandemic has prevented individuals from obtaining necessary reproductive health care [8,9,10,11,12]. Policies and practices enacted in response to the pandemic such as social distancing, stay-at-home orders, and limiting in-person care to “essential” procedures widened gaps in reproductive health access [13], disproportionately affecting underserved populations [8,9,14]. The expansion of telehealth has provided access to reproductive health providers, although barriers still exist [13].

Legislation, policy and education changes over time have enabled pharmacists to expand their scope of practice to address additional societal health needs and challenges [15,16], especially during the COVID-19 pandemic [17]. Community pharmacists have been successful with clinical services including vaccinations [18,19] and smoking cessation [20,21] with services expanding during the pandemic to include COVID-19 testing, vaccination, and prescribing COVID-19 treatments [22]. Prescribing BC would add to pharmacists’ abilities to provide women’s health care and aid in resolving health care gaps caused by public health issues [23]. 

Pharmacist-prescribed BC could increase contraception access to diverse patient populations, improve adherence, reduce unintended pregnancies, and decrease health costs [24,25,26]. In recent studies, women were highly satisfied with pharmacist-prescribed BC, benefited from the convenience of extended pharmacy hours, believed it would reduce unintended pregnancies, and expressed interest in continuing to use pharmacist-prescribed BC [27,28,29,30,31,32]. 

A systematic review of pharmacist perspectives showed that pharmacists were interested, motivated, and comfortable incorporating pharmacist-prescribed BC into their workflow [27]. As of 21 March 2022, 24 US jurisdictions allow pharmacists to prescribe BC by statewide protocol, standing order, or collaborative practice agreement (CPA), but the service has not yet been widely implemented by pharmacists for patients to access. Statewide protocols are preferred as authority comes directly from the state as a ruling allowing all pharmacists to prescribe BC. A standing order is derived by the state department of health director who can grant authority to all pharmacists in the state. Changes in state health directors could result in standing order changes. Additional standing orders were created during the pandemic. With a CPA, pharmacists are granted the ability to prescribe with a contract with a supervising physician. Statewide protocols, standing orders and CPAs can define patients eligible, which contraception products can be prescribed, number of refills, and additional details. Pharmacy boards and licensing bureaus create additional requirements such as training requirements, procedures for prescribing, and reporting if not already addressed in protocols, orders, or CPAs [28]. Some pharmacists with contraception prescribing authority (45%) perceived an increased need for contraception prescribing during the pandemic [33]. However, for those pharmacists actually prescribing, the pharmacy patient volume for BC did not change. 

Exploring pandemic-related effects on contraception is essential. Studies show disasters reduce access to reproductive health care and BC and increase the number of unintended pregnancies, therefore exploring birth control use and access during the pandemic is important [34]. The objectives of this study were to analyze the impact of the COVID-19 pandemic on BC use and access, and evaluate women’s perceptions of pharmacist-prescribed BC during a global pandemic through a survey administered to a convenience sample of women.

## 2. Materials and Methods

### 2.1. Participants

A convenience sample of women aged 18–45 years old was recruited.

### 2.2. Survey Development

A validated survey related to our project was not available. Literature related to emergency contraception, use and safety of hormonal BC available without a prescription, pharmacist-prescribed BC, position statements on pharmacist-prescribed BC, and pandemic impacts on reproductive health and healthcare services were reviewed to create our survey. Survey items were based on published non-validated survey items or newly created. An interdisciplinary team of pharmacists and a medical sociologist developed a 50-item survey with items pertaining to demographics (n = 12); pandemic impact (n = 6); sexual activity (n = 7); BC use and access (n = 17); and perceptions of pharmacist-prescribed BC (n = 8). Some survey questions asked participants about their experience with reproductive health and health care access during three different time periods. Time periods were defined as pre COVID-19 lockdown (1 January–22 March 2020), lockdown (23 March–9 June 2020), and post COVID-19 lockdown (10 June–31 December 2020). A pilot survey was reviewed by ten women with changes made in response to feedback. 

The survey was administered via Qualtrics-XM. Assent was provided before survey completion. Snowballing recruitment technique was encouraged to enroll more participants. Snowballing begins with a convenience sample of initial subjects that recruit a second wave of participants; the second wave participants will then recruit a third wave of participants; and so on. Participants could recruit others by word of mouth or sharing electronically via social media, text, or email. The sample consequently expands by each wave of recruitment [35]. Two Institutional Review Boards deemed the project as exempt.

### 2.3. Community Pharmacy Recruitment and Survey Distribution

Because in-person research was limited during the pandemic, pharmacy staff were utilized to identify participants during face-to-face interactions. Investigators corresponded with corporate chain pharmacy administrators and independent pharmacy managers to secure survey distribution locations. One pharmacy chain (13 stores), 1 grocery store chain (6 stores) and 12 independent pharmacies agreed to participate. The 31 pharmacies were located throughout Michigan to gather opinions from urban, metro, and rural areas. Pharmacy staff were directed to distribute envelopes to patients during walk-in, drive-thru, and medication delivery services. Envelopes contained a study description, survey link, QR code, and a candy bar. A total of 2500 survey envelopes were distributed to the pharmacies, with numbers varying per pharmacy based on rural or urban location, patient volume, and expected response rate. Surveys were distributed from February to April 2021. Pharmacies were contacted at the midpoint and near the end of recruitment to capture survey distribution data. By the end of April, about 2175 surveys were distributed due to one pharmacy having closed and one not distributing surveys.

### 2.4. Electronic Survey Distribution

To increase diversity, the survey link and QR code were also distributed by a Native American Health clinic via their emailed newsletter and posted on an LGBTQ+ support center website’s public domain.

### 2.5. Analysis

Descriptive analyses were conducted with SPSS v27.

## 3. Results

### 3.1. Survey Respondents

A total of 213 survey attempts were received. Surveys were deleted if completed by men or residents outside of Michigan, or less than 50% of the survey answered, yielding 147 acceptable surveys. Participant demographics are in Table 1. Sample age was 29 ± 7.9 years. The majority of respondents were white (77.6%), straight (81.0%), and resided in southeastern Michigan (51.7%). The most common relationship status of respondents was married to a man (38.1%) and living status was with partner/spouse (53.7%). Many respondents (57.8%) used prescription BC with the most common BC formulation being oral pills (76.4%). BC was used for pregnancy prevention (83.7%), menstrual problems (48.8%), acne (24.0%), and polycystic ovarian syndrome (15.1%).

### 3.2. COVID-19 Pandemic Impact

#### 3.2.1. COVID-19 Diagnosis

During the study, access to testing varied dramatically so we also captured perception of COVID-19 illness. During the pre-lockdown period, 4 respondents reported a COVID-19 diagnosis while 16 had COVID-19 like symptoms but no diagnosis. During the lockdown period, 5 respondents reported a COVID-19 diagnosis while 7 had COVID-19 like symptoms but no diagnosis. During the post-lockdown period, 20 respondents reported a COVID-19 diagnosis while 6 had COVID-19 like symptoms but no diagnosis.

#### 3.2.2. Substance Use

Respondents reported an increase in alcohol (29.3%), marijuana (19%), nicotine (10.9%), and opioid (3.4%) use during the pandemic.

#### 3.2.3. Employment

Most respondents reported no change in employment (79%) while some reported being laid off (17.1%) or employed at fewer hours (15.1%).

#### 3.2.4. Insurance

Figure 1 displays the COVID-19 impact on health and prescription insurance. Minimal changes were reported in access to health and prescription insurance. For health insurance, a slight decrease in private and slight increase in public and no insurance occurred during the lockdown period. After lockdown, private health insurance slightly increased while participants insured under parents’ health insurance plans slightly decreased. The number of uninsured respondents remained steady between lockdown and post lockdown timeframes. Fewer changes were seen with prescription insurance coverage except for an increase after lockdown of those without prescription coverage.

#### 3.2.5. Sexual Activity and Pregnancy Worries

Most respondents (79%) were sexually active pre-lockdown, decreasing during lockdown (70.4%) and subsequently increasing post-lockdown (77.4%). Of those that were sexually active pre-lockdown, the mean intercourse activity per week was 2.2 ± 1.6 (range 1–10), decreasing to 2.0 ± 1.9 (range 0–10) during lockdown and post-lockdown 2.1 ± 1.7 (range 0–10). Reasons for changes in sexual activity during lockdown compared to pre-lockdown and post-lockdown can be seen in Table 2. Other reasons for decreased sexual activity during lockdown and post-lockdown included increased workload, lack of childcare, stress, mental health, and other health-related problems. Reasons for increased sexual activity included new relationship, newly married, more privacy, and trying to conceive. 

Of all respondents, 59.0% always used some form of barrier or hormonal BC. Some respondents (13.6%) were worried about becoming pregnant because of a lack of access to barrier or hormonal BC. Some respondents (28.8%) reported being worried about becoming pregnant because of unknown effects the COVID-19 virus could have on the baby.

#### 3.2.6. Prescription BC Use and Access

Of respondents who used prescription BC (n = 85), 15 (17.6%) reported a change in form of BC during the pandemic. Reasons for change included switching to a more convenient form of BC, new medical condition, and/or side effects. Twenty-three (27.1%) respondents reported having difficulty taking BC regularly during the pandemic. 

Twenty-one (24.7%) BC users were worried about BC access. Twenty-one (24.7%) BC users had an appointment for BC canceled, which was rescheduled (n = 12, 14.1%), switched to telehealth (n = 6, 7.1%), or not rescheduled (n = 3, 3.5%). The remainder of the respondents either did not need an appointment or kept their scheduled appointment. In-person appointments for a BC prescription decreased during the lockdown period while telehealth appointments increased (Figure 2). In-person appointments subsequently increased during the post-lockdown period. 

A small number of BC users reported difficulty accessing birth control (n = 7, 8.2%) during the pandemic, with reasons for difficulty including healthcare center being closed, no appointments available, loss of healthcare coverage, unable to afford provider visit, unable to obtain a refill, and unable to afford BC prescription. Three respondents (3.5%) used pharmacist-prescribed birth control during the pandemic.

### 3.3. Perceptions of Pharmacist-Prescribed BC Captured during Pandemic

The likelihood of using pharmacist-prescribed BC by participant characteristics is displayed in Table 1. Because of small sample size, no differences were statistically significant between participant characteristics and the likelihood of using pharmacist-prescribed BC except for reason for BC.

Overall respondents had confidence in the ability of pharmacists to prescribe BC and many reported they would use pharmacist-prescribed BC if it was available to them. (Figure 3). When asked if this service would make it easier to adhere to BC, 27.2% reported definitely yes, 23.5% probably yes, 32.4% maybe, 12.5% probably not, and 4.4% definitely not. Respondents believed that it would be easier to prevent unintended pregnancies if pharmacists had the ability to prescribe BC with 49.3% stating definitely yes, 25.0% probably yes, 16.2% maybe, 8.1% probably not, and 1.4% definitely not. Perceived advantages and concerns of pharmacist-prescribed BC can be found in Table 3.

## 4. Discussion

During the COVID-19 pandemic, some challenges existed with BC use and access. Although a quarter of respondents were worried about access to BC, few respondents reported barriers to obtaining BC. Twenty-seven percent of BC users reported difficulty taking their BC regularly during the pandemic. A quarter of respondents had an appointment for their BC prescription canceled. Overall respondents had confidence in the ability of pharmacists to prescribe BC. Half of the respondents reported they would use pharmacist-prescribed BC if it was available to them. Perceived advantages with receiving a BC prescription from a pharmacist included more convenient, easier to access, time saver, and improved adherence. Some concerns with pharmacist-prescribed BC existed such as not receiving regular PAP smears/screenings, possibility of being prescribed the wrong BC, and encouraging sex at an earlier age.

The current literature on BC access during the pandemic reported challenges with obtaining refills, appointment availability, and affordability, which disproportionately affected minorities and those with financial instability. Our study confirmed emerging data that the pandemic increased worry about BC access [9] and some BC and reproductive health care access barriers existed [9,10,36]. 

We found decreased in-person visits for prescription contraception during the lockdown period with an increase in telehealth visits. Telehealth visits declined post-lockdown but still remained higher than pre-pandemic levels. Telemedicine has been reported to be the most common solution during the pandemic to alleviate access issues [9,36,37]. This trend is consistent with a large national study of public and commercial insurance claims showing a 45% decrease in in-person visits for contraception and a simultaneous 30% increase in telehealth visits from April-June 2020 [37]. Although beneficial, barriers exist for telehealth such as unreliable internet, unsuitable technical devices, and lack of knowledge or comfort with technology [38]. These barriers disproportionately affected historically excluded and underserved communities and sometimes led to patients’ perceptions of receiving inferior care. 

A similar study with 1800 women conducted in Spain in 2020 found changes in sexual activity and BC access during the pandemic. About half of the respondents decreased sexual activity or did not have sex at all during their lockdown period, which could influence BC need. Six percent of women required BC counseling Two percent of the women were able to obtain the counseling over the phone, 2% received it in person, and 2% did not receive counseling at all. Similarly, they also found that the majority of women (93.8%) did not have barriers to obtaining BC. A small number of women (4.4%) experienced barriers to obtaining BC with similar reasons as we found including unable to contact their doctor to renew their prescription or difficulty obtaining refill from the pharmacy [39]. 

Pharmacists, one of the most accessible health care providers [40], are in a suitable position to help increase BC access and alleviate health care access issues. Pre-pandemic studies showed that women would likely use pharmacist-prescribed BC if it was available. A study of Michigan female college students’ opinions on pharmacist-prescribed BC showed high likelihood of using this service with 46.3% reporting extremely likely and 26.3% moderately likely [30]. This study showed slightly higher likelihood to use pharmacist-prescribed BC services compared to ours. The difference could be due to our sample having a wider range of ages and education. We found that ages 18–25 reported higher likelihood to use this service compared to ages 26–31 and 32–45, although the difference was not statistically significant. Women without formal education beyond a high school degree were less likely to use this service but this was not significantly different to those with further education. Our research adds that Michigan women of varying ages and education would likely use pharmacist-prescribed birth control if it was available. 

Although pharmacist prescribing could help alleviate healthcare gaps, barriers and limitations to this service exist. Studies show pharmacists are hesitant to implement this service due to time and resource constraints, lack of reimbursement, and lack of access to medical records [27]. Furthermore, pharmacists are limited to prescribing short-acting BC methods for contraception in most states. The majority of our sample used short-acting hormonal BC for prevention of pregnancy; however, BC was used for other purposes such as acne and menstrual irregularities. Additionally, if a patient prefers a long-acting reversible contraceptive such as an IUD or implant, they would have to seek care from an appropriate provider [41]. Overall, pharmacist prescribing has shown to be a useful service and is well accepted by pharmacists and patients [25,26,27,29,30,31,32]. Further work needs to be done by pharmacists, organizations, and policymakers to advocate and integrate pharmacy prescribing rights into standard pharmacy practice, and help overcome barriers to service implementation.

Because this study was conducted during the pandemic it provides real-time data about public health crises. However, some limitations existed. Recall bias could exist as respondents had to remember events from a year prior. The post-lockdown timeframe is also longer than the other two time frames which could bias results. Additionally, some questions were asked generally during the pandemic without pre/post pandemic data to compare. This study took place during the peak of the pandemic, therefore some pharmacy staff likely had limited time and energy for recruitment efforts. Community pharmacies were adjusting their workflow to accommodate for staffing shortages, personal protective equipment shortages, and implementation of COVID-19 testing. Social distancing measures implemented by pharmacy staff and patients could have limited recruitment. Some community pharmacy patient populations were predominantly older and therefore they could not fulfill recruitment goals. Since some respondents obtained prescriptions from the pharmacy where they received the survey this could lead to bias in responses. This was a long survey resulting in high survey burden with many not completing more than 50% of the survey. Due to the limited sample size, we were unable to detect differences between subgroups. This limited sample size represents a small snapshot of women in southeast Michigan leading to limited generalizability. 

## 5. Conclusions

In a small sample of women in Michigan, the pandemic and resulting lockdowns affected a wide range of factors. There were some changes with BC use and access during the pandemic. Respondents expressed worry about BC access during the pandemic, but only a small number of respondents had difficulty obtaining a BC prescription. Respondents described perceived advantages to pharmacist-prescribed BC, but also reported some concerns. Most respondents have confidence with pharmacist-prescribed BC services. Many respondents reported they would likely use pharmacist-prescribed BC if it was available. BC access issues are important to consider during times of limited provider access and or provider scarcity. Respondents indicated that BC access might be improved with the availability of pharmacist-prescribed BC.

## Figures and Tables

**Figure 1 pharmacy-10-00142-f001:**
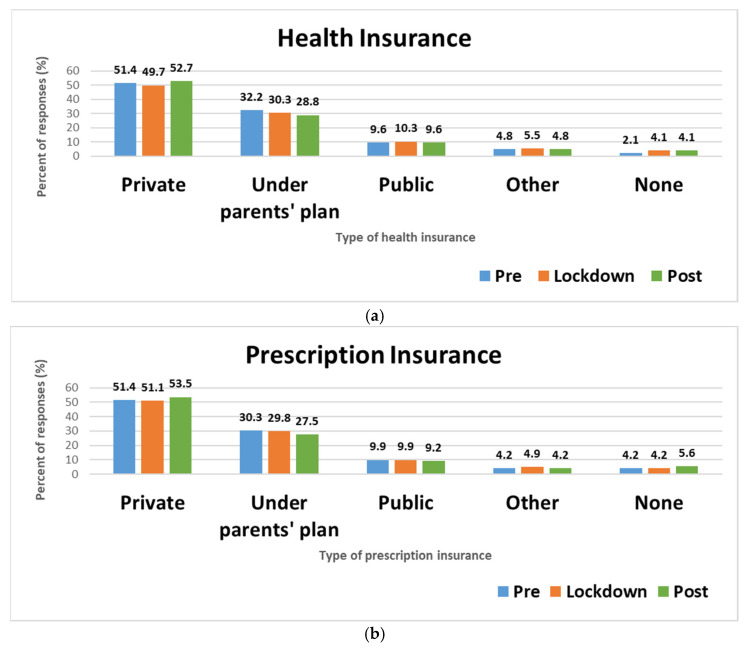
Changes in insurance coverage during pre-lockdown (1 January–22 March 2020), lockdown (23 March–9 June 2020), and post-lockdown (10 June–31 December 2020) periods of COVID-19 pandemic: (**a**) Health insurance; (**b**) Prescription insurance.

**Figure 2 pharmacy-10-00142-f002:**
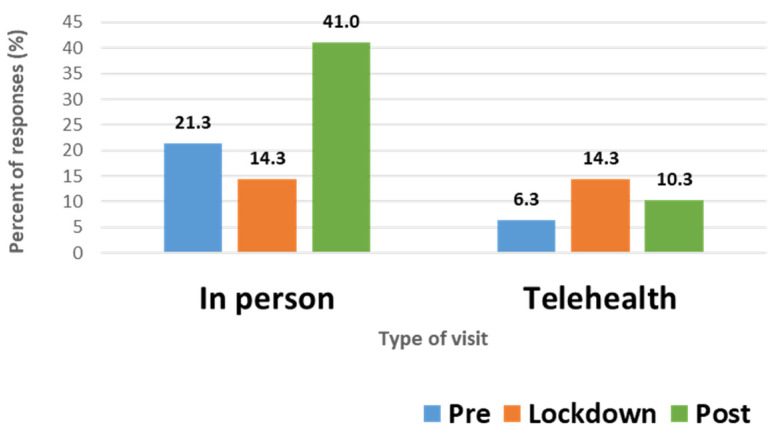
Provider visits for a birth control prescription during COVID-19 pandemic.

**Figure 3 pharmacy-10-00142-f003:**
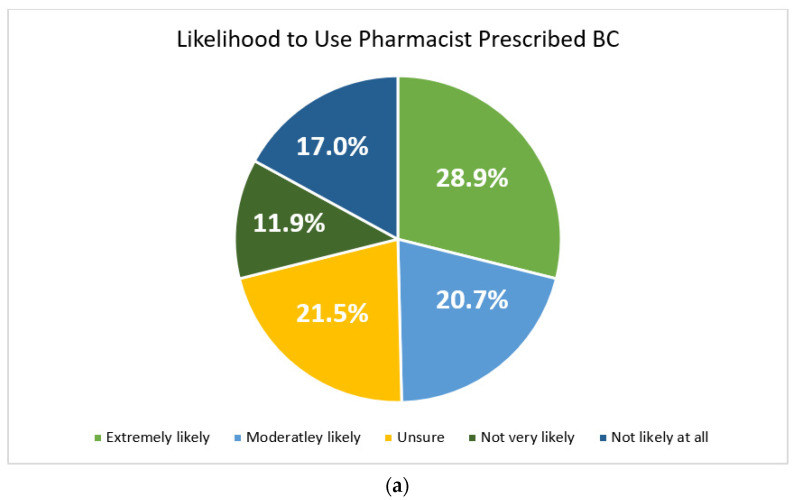
(**a**) Respondents’ likelihood to use pharmacist-prescribed birth control. (**b**) Respondents’ confidence in pharmacists’ ability to prescribe birth control.

**Table 1 pharmacy-10-00142-t001:** Respondent characteristics and likelihood to use pharmacist-prescribed BC evaluated during COVID-19 pandemic.

Characteristic	No. of Respondents n = 147	No. of Respondents Likely to Use Pharmacist-Prescribed BC n = 132–135 ^e,f^	*p* Value ^g^
**Age (years) ^a^**			0.564
**18–25**	49 (34.3%)	26 (57.8%)	
**26–31**	44 (30.8%)	18 (42.9%)	
**32–45**	50 (35.0%)	21 (46.7%)	
**Race**			0.301
**White**	114 (77.6%)	53 (50.5%)	
**More than one race**	15 (10.2%)	5 (38.5%)	
**American Indian**	8 (5.4%)	3 (42.9%)	
**Black**	7 (4.8%)	3 (42.9%)	
**Asian**	3 (2.0%)	3 (100%)	
**Ethnicity**			0.148
**Hispanic/Latinx**	13 (8.8%)	9 (69.2%)	
**Arab/Middle Eastern**	4 (2.7%)	1 (25.0%)	
**Sexual orientation**			0.235
**Heterosexual**	119 (81.0%)	52 (47.7%)	
**LBTQ+**	28 (19.0%)	15 (57.7%)	
**Michigan location**			0.204
**Southeastern**	76 (51.7%)	32 (47.1%)	
**Midwestern**	35 (23.8%)	17 (50.0%)	
**Northern lower peninsula**	18 (12.2%)	11 (68.8%)	
**Upper peninsula**	15 (10.2%)	7 (46.7%)	
**Southwestern**	3 (2.0%)	0 (0.0%)	
**Education**			0.479
**High school or less**	14 (9.5%)	4 (33.3%)	
**Some college/Associate degree**	44 (29.9%)	21 (52.5%)	
**Bachelor’s degree**	54 (36.7%)	25 (50.0%)	
**Graduate degree**	35 (23.8%)	17 (51.5%)	
**Religion ^b^**			0.975
**Christianity**	82 (56.1%)	36 (46.8%)	
**Agnostic**	23 (15.7%)	11 (55.0%)	
**Atheist**	15 (10.3%)	7 (50.0%)	
**Other**	26 (17.8%)	12 (52.2%)	
**Political party**			0.532
**Democrat**	68 (46.3%)	34 (53.1%)	
**Republican**	32 (21.8%)	14 (48.3%)	
**None**	27 (18.4%)	10 (41.7%)	
**Independent**	16 (10.9%)	7 (50.0%)	
**Prefer not to answer**	4 (2.7%)	2 (50.0%)	
**Income**			0.515
**Less than $20,000**	40 (27.2%)	19 (52.8%)	
**$20,000 to $34,999**	27 (18.4%)	12 (48.0%)	
**$35,000 to $49,999**	25 (17.0%)	13 (54.2%)	
**$50,000 to $74,999**	24 (16.3%)	11 (47.8%)	
**$75,000 to $99,999**	7 (4.8%)	2 (28.6%)	
**Over 100,000**	24 (16.3%)	10 (50.0%)	
**Living situation**			0.237
**Living with partner/spouse**	79 (53.7%)	31 (41.9%)	
**Living with parent/guardian**	29 (19.7%)	13 (52.0%)	
**Living on own**	20 (13.6%)	11 (61.1%)	
**Living with roommates**	14 (9.5%)	9 (69.2%)	
**Other**	5 (3.4%)	0 (0.0%)	
**Relationship status**			0.185
**Married to man**	56 (38.1%)	21 (41.2%)	
**Single, long-term relationship**	41 (27.9%)	21 (55.3%)	
**Single, not dating**	22 (15.0%)	11 (55.0%)	
**Single, dating/short term relationship**	19 (12.9%)	12 (66.7%)	
**Other**	9 (6.2%)	2 (25.0%)	
**Use prescription birth control ^c^**			0.136
**Yes**	85 (57.8%)	45 (55.6%)	
**No**	52 (39.1%)	22 (42.3%)	
**BC formulation ^d^**			0.402
**Oral pill**	65 (76.4%)	36 (59.0%)	
**IUD**	15 (17.6%)	6 (40.0%)	
**Other**	5 (5.9%)	3 (50.0%)	
**Reason for birth control ^d^**			
**Prevent pregnancy**	72 (83.7%)	29 (43.9%)	<0.001
**Menstrual problems**	42 (48.8%)	16 (43.2%)	<0.001
**Acne**	21 (24.0%)	7 (38.9%)	<0.001
**PCOS**	13 (15.1%)	8 (66.7%)	0.046

BC, birth control; LBTQ+, lesbian, bisexual, transgender, queer or questioning, plus; IUD, intrauterine device; PCOS, polycystic ovarian syndrome. ^a^ n = 143 due to missing responses. ^b^ n = 146 due to missing responses. ^c^ n = 138 due to missing responses. ^d^ n = 85, sample of respondents who use prescription birth control. ^e^ n = 132–135, total n varies per item due to different numbers of missing responses. ^f^ Percent calculated from row totals in chi square tables; thus within a demographic category, n varies per item subcategory (e.g., age = 18–25 row total n = 45, 28 likely to use pharmacist-prescribed BC 28/45 = 57.8%). ^g^ Comparison between likely, unsure, and not likely to use pharmacist-prescribed birth control per category.

**Table 2 pharmacy-10-00142-t002:** COVID-19 pandemic effects on sexual activity.

	Lockdown Compared to Pre-Lockdown N = 147 (%)	Post-Lockdown Compared to Lockdown N = 136 (%)
Decreased ^a^		
Social distancing	27 (18.4%)	9 (6.6%)
Had COVID-19	6 (4.1%)	4 (2.9%)
No privacy	11 (7.5%)	7 (5.1%)
Other	15 (10.2%)	16 (11.8%)
Increased ^a^		
More time with same partner	26 (17.7%)	23 (16.9%)
More time with multiple partners	7 (4.8%)	7 (5.1%)
Other	0 (0.0%)	8 (5.9%)
No change	59 (40.1%)	63 (46.3%)

^a^ Respondents could choose more than one option.

**Table 3 pharmacy-10-00142-t003:** Perceived advantages and concerns with pharmacist-prescribed birth control evaluated during COVID-19 pandemic.

Advantages	Number Respondents (%); n = 147	Concerns	Number Respondents n = 147 (%)
More convenient	104 (70.7%)	Not receive PAP smears/screenings	89 (60.5%)
Easier access	102 (69.4%)	Prescribed wrong birth control	46 (31.3%)
Save time	99 (67.3%)	Encourage sex earlier	23 (15.6%)
Obtain prescription and fill at same time	96 (65.3%)	Pharmacists lack knowledge	20 (13.6%)
More accessible hours	88 (59.9%)	Pharmacists lack skills	15 (10.2%)
Less likely to run out of birth control	81 (55.1%)		
Less costly than provider visit	80 (54.4%)		
Greater confidentiality	26 (17.7%)		

## Data Availability

Not applicable.

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
