# Peer review of "Birth Control Use and Access Including Pharmacist-Prescribed Contraception Services during COVID-19"

_pharmacy, 2022, doi:10.3390/pharmacy10060142_

Round 1
Reviewer 1 Report
The authors attempted to assess how the covid-19 pandemic affected access to contraceptive methods. One of the main objectives of the study was to evaluate the perception of fertile women (18-45) on the possibility of pharmacists prescribing BC.
First of all, the text must be formatted according to the journal's requirements (e.g. justified text, remove space after paragraph).
Introduction - a more detailed description of the law that allows pharmacists to prescribe contraceptives is needed. Is it only about refills, or can they prescribe hormonal products without a medical evaluation? (in the case of hormonal products the risks must be evaluated by a physician). Readers from other countries are not aware of this law.
Line 36 – a point (.) is missing after [3,4]
Material and methods
Line 80 - It should be stated how many women completed the survey.
Did the authors calculate the sample size to be representative for Michigan? Which method was used in determining the sample size?
Is there an explanation for the low responsiveness rate?
Line 93 – please delete ]
In table no. 1:
- age total no. is 143 (not 147) please correct
- religion 146 (not 147) please correct
Discussion – try not to use our study so often at the beginning of sentences.
Some references to the limitations in prescribing contraceptives by pharmacists are needed. Even within the study, the authors identified that some of the respondents used these products for the treatment of acne, menstrual disorders or PCOS, which require diagnosis and are not within the competence of the pharmacist to treat.
I believe that the study was not properly designed from the beginning, however I appreciate the authors' attempt to draw some conclusions even if the response rate was low.
Reviewer 2 Report
See attached File.

Reviewer 3 Report
Overall Impression: This is a well-written but poorly focused manuscript describing the results of a rather extensive survey administered to individuals. The response rate for usable surveys was quite low at < 7%, which was undoubtedly influenced by the length of the survey and possibly the intimacy of some of the questions.
It purports to “analyze the impact of COVID-19 pandemic on BC use, adherence, and access; and evaluate women’s perceptions of pharmacist-prescribed BC during a global pandemic,” yet adherence is never addressed. Additionally, it includes data on seemingly unrelated topics, such as COVID-19 illness, substance use, and utilization of non-related pharmacy services such as 90-day supplies, vaccination, and curbside pick-up.
A major criticism of this study is its low response rate, putting it at risk for bias. In addition, results that are stratified by time period are inherently biased due to significantly different period of time for each (pre-COVID =81 days, during lockdown=78 days, post-lockdown=204 days).
The authors clearly put a lot of work into creating, administering, and compiling this survey, but it is not clear what this adds to the literature that has not already been reported. The authors own cited references present studies with very similar results done with larger sample sizes.
Introduction:
· Specific Comments:
o 44-45: This is statement makes it seem that there is a causal relationship between infectious disease breakouts and their effect on productive health services. Would recommend rewording as this is not something that seems to have been proven.
o 47: Telehealth is said to have “widened gaps in reproductive health access,” and then in line 49-50 to have “provided access to reproductive health providers” These statements are contradictory.
o 57: “prevention medication prescribing” What preventative measures you are referring to? COVID-19 prevention? HIV-PrEP prevention?
o 74: Why is exploring pandemic-related effects on contraception essential? Include more information about why this study was performed and why it is important to literature.
Methods:
· The variability of the three time periods introduces bias towards the longer time frames. The post-period is about twice as long as the pre- and during-, so it is not surprising that there were more in person healthcare visits (Figure 2) during this time.
· The surveys were distributed by different methods—through the letters handed out at pharmacies, the Native American Health clinic newsletter, and the LGBTQ+ website-- how many people responded within each distribution method?
· Specific Comments:
o 80: Which pharmacies? What type of pharmacies? This is listed later in the section, but should be here
o 83: over-the-counter birth control – could you expand on this? This does not seem to be legal in Colorado
o 96: “Snowballing recruitment” should be briefly explained and referenced.
Results:
· Overall Comments:
o Would include response rate to survey; how many patients started but did not complete at least 50% of the survey?
o About 40% of patients who responded that they would use pharmacist prescribed birth control are on IUDs, was this considered when presenting results? Were patients made aware what types of birth control they would be able to receive from pharmacists vs physician?
o How does substance abuse, sexual activity, and utilization of non-related pharmacy services such as 90-day supplies, vaccination, and curbside pick-up (Table 3) contribute to your objectives? Some of these results seem un-related.
· Specific Comments:
o 133 (Table 1): na= 132=135%, it is not clear what this percentage is referring to
§ Also, the percentages do not add up in the ages section for the No. of respondents (%): 33.3 + 39.9 + 34.0 = 107%
· Would ensure all percentages are correct
§ Michigan location such as “Southeastern, Midwestern, Northern lower peninsula…” is not very meaningful to anyone outside of Michigan; would be more generalizable to describe in terms such as urban vs rural
Discussion:
· Overall Comments:
o Results should not be repeated, rather highlighted, and put into context with results from similar studies.
· Specific Comments:
o 272: First report of results related to adherence rates. This should go in the results section.
o 308: “From comparison of these two studies, we can conclude…” No formal comparison of the two studies is presented; and this is inaccurate based on the results presented in line 258, where 28.9+20.7% (=49.5%) of women were extremely or moderately likely to use pharmacist-prescribed BC
o 319-320: awkward sentence
Conclusion:
· Overall Comments:
o Would ensure that your main findings and main objectives (lines75-76) are concluded in this paragraph vs. topics or results that were not discussed as part of the main focuses of the research.
· Specific Comments:
o 332: Which question was asked specifically in the survey that would allow for this statement to be true?
References:
Would ensure all are appropriately cited and have links that are correct, as some links had the wrong DOI—opening to a different article than the one cited.
Round 2
Reviewer 1 Report
The manuscript has improved after revision, but I still suggest some comments for minor revisions.
- images quality is poor
- line 304 – Table 34? Please correct with table 3.
As I stated in the first report, there were some methodological errors from the beginning.
Author Response
- images quality is poor. We tried to improve image quality to the best of our abilities.
- line 304 – Table 34? Please correct with table 3. Corrected
Reviewer 3 Report
This is definitely improved, including clarification and narrowing of the scope. A few minor suggestions are in the attached document
